# Can Large Language Models Generalize Procedures Across Representations?

Fangru Lin [1]   Valentin Hofmann [2][3]   Xingchen Wan [4]   Weixing Wang [5]   Zifeng Ding [6]   Anthony G. Cohn [7][8]
Janet B. Pierrehumbert [1]

## Abstract

Large language models (LLMs) are trained and tested extensively on symbolic representations such as code and graphs, yet real-world user tasks are often specified in natural language. To what extent can LLMs generalize across these representations? Here, we approach this question by studying isomorphic tasks involving procedures represented in code, graphs, and natural language (e.g., scheduling steps in planning). We find that training LLMs with popular post-training methods on graphs or code data alone does not reliably generalize to corresponding natural language tasks, while training solely on natural language can lead to inefficient performance gains. To address this gap, we propose a two-stage reinforcement learning curriculum that first trains on symbolic, then natural language data. The curriculum substantially improves model performance across model families and tasks. Remarkably, a 1.5B Qwen model trained by our method can closely match zero-shot GPT-4o in naturalistic planning. Finally, our analysis suggests that successful cross-representation generalization can be interpreted as a form of generative analogy, which our curriculum effectively encourages. The dataset and code used in this paper can be found here.

## 1. Introduction

While humans use natural language as a primary way to reason and communicate about goals and procedures (Carberry, 1990; Lupyan, 2016) (e.g., describing ordered steps for a planning task), large language models (LLMs) are extensively trained on symbolic data such as code and graphs

[1]University of Oxford [2]LMU Munich [3]Munich Center for Machine Learning [4]Google Cloud AI Research [5]Hasso Plattner Institute [6]University of Cambridge [7]University of Leeds [8]The Alan Turing Institute. Correspondence to: Fangru Lin <fangru.lin@ling-phil.ox.ac.uk>.

*Proceedings of the $43^{rd}$ International Conference on Machine Learning*, Seoul, South Korea. PMLR 306, 2026. Copyright 2026 by the author(s).

(Muennighoff et al., 2023; Aryabumi et al., 2024; Ye et al., 2024). It is important to understand whether and how LLMs can use procedures learned in symbolic representations to solve natural language problems, as real-world tasks are often specified in natural language. Success in such a setting requires efficient **cross-representation generalization**, a crucial but underexplored capability for robust LLMs. While existing work studies the effect of various symbolic training on natural language reasoning (e.g., Petty et al. 2024; Zhang et al. 2024; Li et al. 2025), they often conflate surface representational factors and deeper structural ones, providing limited insights into why symbolic training helps in some cases but hurts in others.

Cross-representation learning is considered a form of **generative analogy** in cognitive science (Gentner, 1983; Falkenhainer et al., 1989). Such analogies enable humans to map learned structures across representations in a zero-shot manner by identifying and aligning shared underlying structures (Hummel & Holyoak, 1997; Doumas et al., 2008; 2022; Fang & Sims, 2025). As a result, generative analogy plays a central role in human reasoning and novel discovery. Existing works on LLMs primarily focus on *proportional analogies* about simple surface-level relations, such as *king* is to *man* as *queen* is to *woman* (Mikolov et al., 2013; Yuan et al., 2023b; Petersen & van der Plas, 2023; Yang et al., 2025), while deeper structural transfer across representations is relatively less well understood (Yuan et al., 2023a; Hofmann et al., 2024; Sultan et al., 2024).

In this work, we study cross-representation generalization in a carefully controlled setting where task data is isomorphically presented in three representations: natural language, graph, and code. By varying surface forms but maintaining underlying algorithms, we isolate procedure generalization from spurious transfer. We mainly report Qwen model (Yang et al., 2024) performance on the challenging task of asynchronous planning (Figure 1), which has high ecological validity due to the realistic nature of its data source and has received growing attention recently (Lin et al., 2024a; Ding et al., 2025; Wei et al., 2025). To examine the generalizability of our conclusion, we further evaluate our findings on two additional tasks (maths and physics (Huang et al., 2025)) and model families (Llama-3 (Dubey et al., 2024) and Olmo-2 (OLMo et al., 2024)). We experiment with four

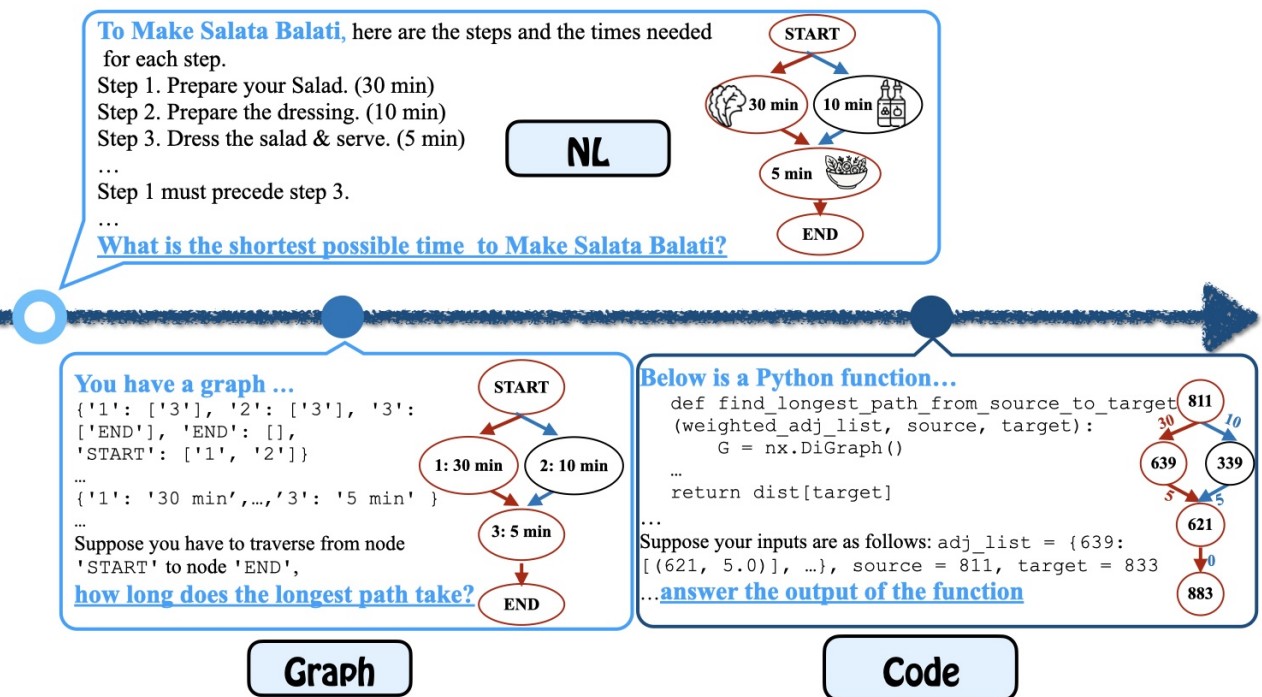

*Figure 1.* Illustration of cross-representation generalization of the planning task studied in this paper. Each frame describes a question and its corresponding underlying graph problems as well as their solutions (critical paths in **red**). The top frame describes a natural language planning problem in NL, where the problem is given in natural language. Essentially, the shortest time needed for this task can be solved by formalizing the constraints of the problem as a DAG and calculating the longest directed path. The bottom part describes two other proxies, namely Graph and Code. Both proxies share exactly the same procedures as NL and can be solved by the same algorithms. The only difference is the representation format. See detailed prompt illustrations in Appendix A.1.

supervised fine-tuning (SFT) and reinforcement learning (RL) methods, namely, vanilla SFT, distillation (Gu et al., 2024), Self-Taught Reasoner (STaR; Zelikman et al., 2022), and Group Relative Policy Optimization (GRPO; Shao et al., 2024). We find that **none of these post-training methods exhibit systematic structure mapping in naturalistic data when trained only on symbolic data** across LLM families and scales. This shows that LLMs do not naturally generalize learned procedures across representations.

To bridge this gap, **we propose a two-stage reinforcement learning curriculum of first training on symbolic, then on naturalistic data.** We show that using this curriculum, Qwen2.5-1.5B-Instruct outperforms its 7B variant under GRPO and is even on par with zero-shot GPT-4o[1]. This curriculum is also more robust in input variations and shows consistent gains across model families and tasks.

Last, we analyze when and how LLMs' cross-representation generalization emerges. We show that successful cross-representation generalization can be interpreted as a form of generative analogy rather than frequency-based learning, which our curriculum effectively encourages. Overall, our results demonstrate that while LLMs can generalize across

representations via an appropriate curriculum, they require extended training in the new representations. This stands in contrast with human analogical behaviors, which exhibit cross-representation generalization with minimal exposure.

## 2. Asynchronous Planning in Natural Language, Code, and Graph

In this section, we describe our primary task, asynchronous planning in natural language (NL), and how we build Graph and Code data procedurally equivalent to NL to study cross-representation generalization. We focus on naturalistic asynchronous planning problems introduced in AsyncHow (Lin et al., 2024a; Figure 1 top, NL). Each data point in AsyncHow describes a task (e.g., making Salata Balati) with relevant steps, step time duration, and step dependencies, where some steps can be parallelized. The model tested is required to give the shortest time possible to complete the task, assuming infinite resources are available.

As illustrated in Lin et al. (2024a), each planning instance can be formalized as a Directed Acyclic Graph (DAG), where nodes correspond to steps and edges encode dependency constraints. The time estimation task is equivalent to computing the time duration of the critical path in the

---

[1]https://openai.com/index/gpt-4o-system-card/

DAG (Figure 1 top right). The procedural equivalence of naturalistic and symbolic representations of this task makes it a natural testbed to study structural generalization across representations, as we can substantially vary data surface form but keep underlying structures unchanged.

Inspired by recent works using graphs and code during training and test time (Ye et al., 2024; Li et al., 2025; Wang et al., 2023; Gao et al., 2023; La Malfa et al., 2025), we build `Graph` and `Code` as two representation-wise distinct but procedurally equivalent proxies for `NL` (Figure 1 bottom, full prompts in Appendix A.1).

**Graph.** (Figure 1, bottom left) We build an adjacency list representation as a dictionary for each implicit dependency graph in a natural language question `NL` by translating each step constraint as a graph property (e.g., `Step 1 must precede step 3` is represented as `'1':['3']` in the adjacency list). We provide two dummy nodes `'START'` and `'END'` for full connectivity. We also encode time constraints for each node in a separate dictionary (e.g., `{'1': '30 min', ...}`). We ask the tested model to compute the duration of the longest path, where dummy nodes take no time to traverse.

**Code.** (Figure 1, bottom right) We present a Python code snippet that implements a DAG longest-path search algorithm. We randomly assign different indices to nodes, then convert all time descriptions to numeric values in minutes (e.g., 1 hour is rewritten as 60) and use them as weights for edges connecting these nodes. At test time, the model is provided with: (i) the Python function implementation, (ii) an adjacency list of nodes and weighted edges, (iii) the start and end nodes. Finally, we ask the model to give the output of the function for the given input.

## 3. Main Experiment

We ask the following questions in our main experiment.

1. Can procedural knowledge learned from symbolic representations (`Code` or `Graph`) transfer to natural language problem solving (`NL`)?

2. How do model scale and post-training methods affect cross-representation generalization?

### 3.1. Experimental Settings

In this subsection, we describe the setting of the main experiment. We train three families of models with four different post-training strategies. For each base model and training method, we train on exactly one representation of the same underlying task (`NL`, `Graph`, `Code`), then test on all three. Since the underlying structures are isomorphic across representations, the performance difference under the representation shift can directly reflect how well procedures generalize.

**Base Models.** We mainly use Qwen-2.5-Instruct (1.5/3/7B; Yang et al., 2024) as our base model since it demonstrates superior performance compared to models of similar or even larger scales and is used in DeepSeek-R1 as a base model for training (DeepSeek-AI, 2025). We study multiple model sizes to examine the scaling effects with a reasonable computational budget.[2]

To assess the generality of our observation, we additionally perform experiments on Llama-3.2-1/3B-Instruct, Llama-3.1-8B-Instruct (Dubey et al., 2024), and Olmo-2-1/7B-Instruct (OLMo et al., 2024). For clarity, we present Qwen results in the main content, and defer additional results on Llama and Olmo to Appendix A.2, where we see similar trends.

**Post-training methods and implementation details.** We use four popular post-training approaches: three SFT methods and one RL method.[3]

**SFT methods.** We use three SFT variants in the experiment.

1. **Vanilla SFT.** We use this method as a baseline, where we finetune the model only on the prompt and the answer without chain of thought supervision (CoT; Wei et al., 2022b; Kojima et al., 2022).

2. **Distillation.** We use DeepSeek-R1-Distill-Qwen-32B as the teacher model to generate CoT reasoning responses for student model training. We initialize with temperature $= 1$ and generate $k = 4$ responses per prompt for diversity. We retain only the responses whose answers exactly match the ground truth final answers as distillation data. For prompts without correct answers, we randomly sample 5 prompt–answer pairs from the correct generations as in-context learning examples (Brown et al., 2020). We repeat for 5 rounds with increasing sample sizes, resulting in a high-quality training set (details in Appendix A.3.1).

3. **STaR.** We perform a 10-iteration bootstrapping process using Self-Taught Reasoner (STaR; Zelikman et al., 2022), where the model is progressively trained on its own generations. In each iteration, we sample $k = 4$ generations per prompt with temperature $= 1$ for diversity. We retain the generations whose answers exactly match the ground truth final answers, and randomly

---

[2]We do not perform experiments on larger-scale models because they perform reasonably well on this task without the need for further tuning.

[3]To save compute, we follow prior work and do not vary the RL method (cf. Chu et al., 2025).

sample one correct response per prompt. The base model is then fine-tuned only on the filtered subset with correct answers. We then sample a new training set from the fine-tuned model, and train the base model again on this new training data. To obtain a larger training set, we also use rationalization introduced in Zelikman et al. (2022), providing ground truth answers as hints for the prompts unable to lead to qualified responses during generation.

**RL method.** We use Group Relative Policy Optimization (GRPO; Shao et al., 2024), the primary RL method used in DeepSeek-R1 (DeepSeek-AI, 2025). We sample $k = 16$ generations for each prompt and train with verifiable outcome rewards (Lambert et al., 2024), where answers that are both correct and adhering to the required format get a reward of 1, otherwise 0.

All training methods update full parameters. SFT models are trained for 2 epochs. GRPO models are trained for one epoch, 20 episodes. All SFT methods are implemented based on Llama Factory (Zheng et al., 2024), and GRPO is based on OpenRLHF (Hu et al., 2024), with hyperparameters adopted from Open-Reasoner-zero (Hu et al., 2025). See Appendix A.3 for more implementation details.

**Training and test data.** To evaluate generalization across representations, we first train base models on a single representation in NL, Graph, Code, then test on all representations of asynchronous planning (results on math and physics are in Appendix A.2). We split each dataset into training and test sets following Lin et al. (2024b), with stratified sampling based on complexities defined in Section 2. After deduplication, each training set has 1,364 and the test set has 225 data points.

Our primary target is NL, which reflects practical usage in natural language interfaces. Following Zhang et al. (2024), we consider a model to have *significant transfer* if the performance improvement after training is statistically significant compared with an untuned baseline. We use McNemar's tests (McNemar, 1947) as the significance measure. Models that acquire generalizable procedural knowledge should be able to have *significant transfer* and strong performance in unseen representations.

### 3.2. Results

We report post-training accuracy in Figure 2, and the performance delta compared to untuned baselines in Appendix A.5. We discuss our main findings below.

**Overall failure of cross-representation generalization. Across all LLMs and post-training methods, naive training on a single representation fails to reliably generalize across representations**. While fine-tuning methods uni-

formly offer a strong within-representation advantage, their performance often collapses under representation shifts, although tested instances share identical underlying procedures. Even when a setting exhibits statistically significant transfer, out-of-representation performance remains markedly lower than within-representation performance (e.g., training Qwen-2.5-Instruct-1.5B on Graph using STaR), indicating that learned knowledge is incomplete and weak. This observation extends findings on in-context analogical string manipulation (Lewis & Mitchell, 2024; Qin et al., 2024) to parametric post-training on natural tasks.

**Comparison of training representations.** Training on within-representation NL yields the strongest performance on the NL test data. However, small models still struggle even with within-representation training (around 0.5 accuracy for 1.5/3B scales). Training on Graph and Code tends to achieve high within-representation performance, but transfers poorly to NL. This raises concerns about relying only on symbolic data (e.g., Li et al., 2025) to improve the performance on complex natural-language user queries.

**Comparison of different post-training methods.** GRPO (RL method) has the strongest within-representation performance across tested methods. However, its relative advantage diminishes under representation shift, despite significantly more training compute. This finding questions whether RL can genuinely learn generalizable procedures in its current paradigm, or simply exhibits better shallow pattern exploitation (Cobbe et al., 2020).

Among SFT methods, vanilla SFT performs the worst overall. Distillation substantially improves within-representation performance, approaching GRPO with much less training budget, but it sometimes exacerbates cross-representation degradation (see details in Figure 6). Finally, STaR, as a self-bootstrapping method, demonstrates stronger generalization performance compared to vanilla SFT, though it remains weaker than distillation without the benefits of a more capable teacher model.

**Comparison of model sizes, families, and tasks.** Scaling model parameters does not qualitatively change cross-representation procedure generalization patterns. Although larger models tend to have a more noticeable performance drop in unseen representations, it is likely due to their stronger performance on the trained representation rather than weaker transfer capabilities.

Overall, these results suggest that current popular post-training methods often learn task-specific surface patterns rather than representation-agnostic procedures, and simply scaling model sizes does not provide visible benefits. Our observation is consistent across different model families and extends to math and physics tasks (details in Appendix A.2).

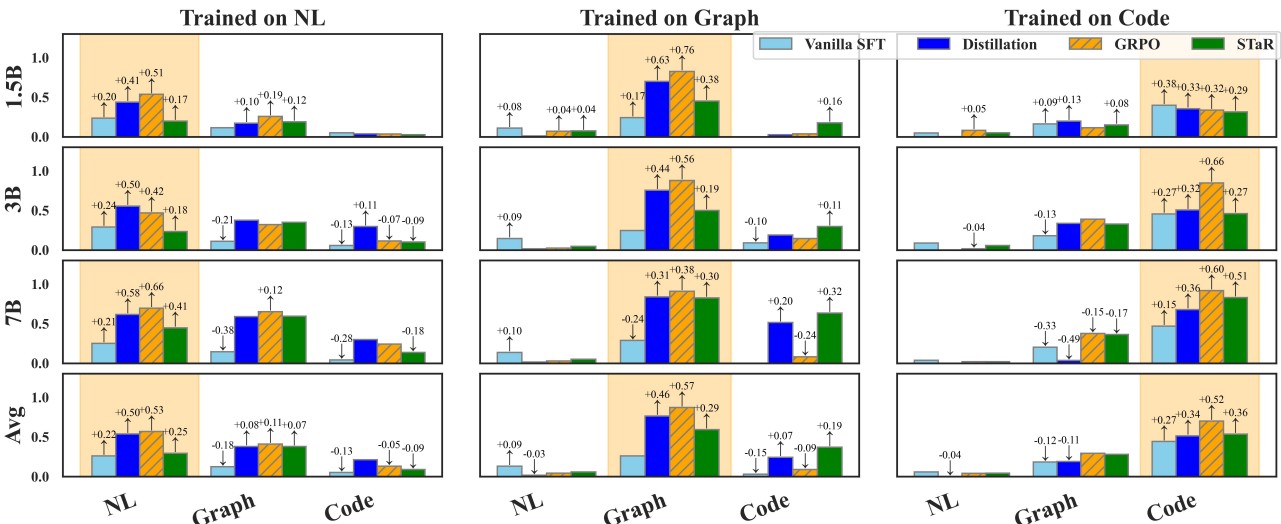

*Figure 2.* Qwen models' performance after training. We train models on one representation from NL, Graph, and Code, respectively (left to right), and test them on all representations (with the particular aim to optimize on NL). Results in **orange** background are in the same distribution as the training set (e.g., train on NL and test on NL). Up/down arrows denote test results significantly better/worse than untuned baselines by McNemar's tests (McNemar, 1947), and numbers by the arrows indicate performance delta. Models generally cannot transfer learned procedures in Code and Graph to NL, despite showing high within-representation performance (delta results in Appendix A.5).

## 4. Curriculum for Cross-Representation Generalization

In Section 3, we observe that popular post-training methods often fail to generalize procedural knowledge across representations. Training on symbolic data only transfers weakly (or not at all) to natural language, while training on natural language alone can yield much slower performance growth. To address this gap, we propose a two-stage curriculum to strengthen cross-representation generalization. Essentially, we separate learning into two stages: (i) symbolic induction, where the model is trained on symbolic data (e.g., Code or Graph) to learn abstract procedures, then (ii) natural language adaptation, where the model is continually trained on NL so the procedures become usable in natural language.

### 4.1. Main Results

We evaluate our curriculum by training a 1.5B Qwen model with GRPO. In stage 1, we train on Graph for 40 steps (20 episodes). In stage 2, the training continues for another 40 steps in NL. As a baseline, we train an identical model solely on NL for 80 steps. Since we fix the total training steps and sequence length cut, each condition is essentially controlled with the same training token budget.

Results are in Table 1. We find that, with the same training budget, our curriculum substantially outperforms NL-only training, even though it has fewer within-representation NL updates. The reward curve of the stage 2 curriculum learning resembles the symbolic training dynamics more than direct

NL training (Appendix A.4). This might indicate that the stage 1 symbolic warm-up in our curriculum is helpful in altering dynamics in NL training, encouraging generalizable procedure learning.

The curriculum also provides an efficient scaling benefit: the 1.5B model trained for 80 steps outperforms the 3B model trained on NL alone for 40 steps (same budget). It also outperforms its 7B variant on NL, which requires approximately ×2.3 as much budget for training. This curriculum also enables our 1.5B model to outperform zero-shot GPT-4o-mini and match GPT-4o on the asynchronous planning task.

We note that **the curriculum order matters**: reversing the order of the curriculum to NL→Graph performs markedly worse than NL-only training (0.431 vs. 0.698 test accuracy on NL). **The training method also matters**: replacing stage 2 GRPO with NL distillation, the best SFT method observed in previous subsections, leads to much worse results (0.462 test accuracy on NL). This curriculum is not effective in standard SFT either: (0.236 acc on NL-only, 2 epochs vs. 0.244 acc on NL-only, 4 epochs vs. 0.249 on Graph 2 epochs→NL 2 epochs).[4] This indicates that RL can be substantially more effective than SFT in cross-representation generalization.

Finally, we validate the same curriculum on Olmo-2-7B-

---

[4]Similarly, we observe no meaningful improvement in parameter-efficient SFT methods such as rank 8/16 LoRA (Hu et al., 2022).

*Table 1.* Training and test accuracy on `NL` and `NL-AAVE` datasets for various training strategies and baseline models. Best results are in **bold**. Second-best results are underlined. Results with * are a statistically significant improvement over 80-step `NL` by McNemar's tests (McNemar, 1947).

| Train Setting | Train Acc. (NL) | Test Acc. (NL) | Test Acc. (NL-AAVE) |
|---|---|---|---|
| Graph (40 steps) + NL (40 steps) | **0.873**$^*$ | **0.782**$^*$ | 0.573 |
| NL (40 steps) + Graph (40 steps) | 0.504 | 0.431 | 0.169 |
| NL only (80 steps) | 0.811 | 0.698 | 0.507 |
| *Zero-shot and Baselines* | | | |
|    3B (NL 40 steps) | 0.556 | 0.471 | 0.400 |
|    7B (NL 40 steps) | 0.753 | 0.698 | 0.573 |
|    GPT-4o-mini (zero-shot) | – | 0.440 | 0.289 |
|    GPT-4o (zero-shot) | – | **0.782** | **0.724** |

*Table 2.* Accuracy of models trained on different data representations. Boxes in yellow are in-domain results. Best results are in **bold**.

| | Train setting | Test Acc. (Math) | Test Acc. (Physics) |
|---|---|---|---|
| Math | Code (20 steps) + NL (20 steps) | **0.435** | **0.230** |
| | NL (40 steps) | 0.385 | 0.135 |
| Physics | Code (20 steps) + NL (20 steps) | **0.325** | 0.550 |
| | NL (40 steps) | 0.315 | **0.555** |

Instruct, which exhibits a similar gap between symbolic and naturalistic training. We observe consistent performance gains, indicating that **our curriculum generalizes across model families** (detailed results in Appendix A.6).

### 4.2. Ablation on Curriculum and Data Mixture

**Ablating curriculum.** We perform additional experiments on interleaved `Graph+NL` training (i.e., using `Graph+NL` training data together for 40 gradient steps). While we control the same training budget, we find that the interleaved training underperforms `Graph→NL` curriculum (acc 0.382 vs. 0.782 on NL). This observation is in line with existing literature, which shows that learning multiple tasks at the same time can result in task interference and that curriculum learning can outperform joint training (Pentina et al., 2015; Standley et al., 2020).

**Varying Data Mixture.** The main inspiration for the `Graph→NL` curriculum is that some representations (e.g., `Graph`) train much faster than `NL`. However, `Code` does not show such an advantage (0.338 acc for 40 steps). `Code (40 steps)→NL (40 steps)` is even worse than `NL`-only (40 steps) (0.382 vs. 0.538 acc). Similarly, `Graph+Code→NL` is worse than `Graph→NL` (0.533 vs. 0.782 acc), because the first phase is too weak (0.522 acc). Moreover, we note that `Graph (40 steps)→Code (40 steps)` (0.787 acc) is more effective than `Code`-only (80 steps) (0.373 acc) when tested in `Code`. These results indicate that building a strong foundation in the first

step is crucial for the success of inductive bias in the final task stage.

### 4.3. Robustness to Linguistic Variation

To test the robustness of our curriculum to variations, we further introduce `NL-AAVE`, a dialect variant written in African American Vernacular English (Lin et al., 2024b). `NL-AAVE` preserves the task structure of `NL` but changes surface form in a way that is even challenging to frontier models which excel in the `NL` planning task (Lin et al., 2024b).

Our results in Table 1 suggest that the curriculum consistently outperforms the baseline methods (`NL`-only and `NL→Graph`) on `NL-AAVE`. It also outperforms GPT-4o-mini and the 3B Qwen model trained on `NL` only. This is remarkable considering that the curriculum-trained model does not have explicit exposure to the dialect during training, supporting the interpretation that our curriculum learns generalizable procedures rather than memorizing surface-level cues.

### 4.4. Curriculum in Math and Physics

We further assess if the same idea extends beyond our core planning domain by conducting analogous experiments in physics and math. Specifically, we focus on hard math questions from levels 4 and 5 from MATH (Hendrycks et al., 2021) and physics questions from SciBench (Wang et al., 2024). We use Python parallel data (i.e., `Code`) from

Huang et al. (2025) for stage 1, followed by a naturalistic adaptation phase on `NL`. We split the math dataset using stratified sampling by complexity, reserving 200 instances for the test set and the remaining for training (200 test and 1411 train instances in total). For physics, we also use 200 instances for testing and leave 227 instances for training. Due to constrained compute, we compare Qwen2.5-1.5B-Instruct trained with GRPO on `NL` only for 40 steps versus a curriculum of `Code` for 20 steps followed by `NL` for 20 steps.

We report results in Table 2. Generally, even when our curriculum shows comparable or even slightly lower in-domain performance, it generalizes better across domains (e.g., training on math with a curriculum generalizes better to physics test data). This indicates that our method is helpful in learning generalizable representations.

# 5. Analysis

After establishing that existing training methods yield limited generalization, while our proposed curriculum substantially improves performance, we ask how LLMs generalize procedures and why our curriculum works. In this section, we show that successful generalization can be interpreted as a form of generative analogy rather than pure frequency effects, and that our curriculum effectively encourages the analogical behavior. We also provide qualitative examples of how different training settings affect the learned procedures.

## 5.1. Analogy or Frequency-based Generalization?

A core question behind generalization across representations is whether success is primarily driven by data frequency or structural similarity. Consider a planning task whose critical path is not obvious: a model can succeed either by composing knowledge from many moderately similar instances or by directly mapping from a few highly similar instances in the learned data. These two approaches feature frequency-based and analogy-based learning. Understanding LLMs' generalization patterns in these regards helps us explain both the limited transfer from standard learning and why our curriculum works better.

Following Hofmann et al. (2024), we examine two hypotheses for generation patterns: frequency- and analogy-based hypotheses. *Frequency-based hypothesis* attributes success to the exposure to a sufficiently large *quantity* of relevant training instances. *Analogy-based hypothesis* emphasizes the structural similarity of the most relevant item, rather than their sheer frequency.[5]

**Analogical strength.** We quantify the similarity between two items by *analogical strength* following structure-mapping theory (Gentner, 1983). This measure quantifies the structural similarity between the train and test instances in a representation-agnostic way by leveraging the underlying representations as DAGs. We denote the base data and target data, which can vary in surface representation forms as $D_b, D_t$, respectively. Three guiding constraints from Gentner (1983) are *structural consistency* (one-to-one mapping), *parallel connectivity* (connected predicates in the base data should also be connected in the target data), and *systematicity* (deeper relational mappings contribute more than shallower ones).

The underlying structure of our stimuli can be expressed as DAG $G = \langle V, E, w \rangle$, where $V$ and $w$ are unary and $E$ is binary (i.e., it connects two predicates). The analogical strength between a base data DAG $G_b = \langle V_b, E_b, w_b \rangle$ and a target data DAG $G_t = \langle V_t, E_t, w_t \rangle$ by weighting the similarities of items can be described as follows:

$$AS(G_b, G_t) = \alpha \Big( \underbrace{sim_u(V_b, V_t) + sim_u(w_b, w_t)}_{\text{unary item similarity}} \Big)$$
$$+ (1 - \alpha) \Big( \underbrace{sim_b(E_b, E_t)}_{\text{binary item similarity}} \Big), \quad \alpha < 0.5 \tag{1}$$

$sim_u$ and $sim_b$ are functions that quantify the similarity between items of different relational complexities. $\alpha$ is a discount factor to assign more weights to binary items than unary items. In practice, we set $\alpha = 0.4$, and measure unary item similarity by multi-set histogram-Jaccard over the distribution of node time durations. We measure binary item similarity by Weisfeiler–Lehman subtree kernel (Shervashidze et al., 2011) with 3 iterations over task graphs.[6]

Intuitively, if successful instances tend to appear with many training items with moderate analogical strength, the generalization corresponds to a frequency-based explanation. If success is more correlated with training instances of high analogical strength, it suggests that the generalization adheres more to an analogical behavior pattern.

**Correlation analysis.** We focus on GRPO-trained Qwen-2.5-1.5B-Instruct on `NL`/`Graph`-only training, and `Graph`→`NL` curriculum. We restrict our analysis to the train items that the model can answer correctly after training. We evaluate the hypotheses by correlating test accuracy with (i) the count of successfully learned training items beyond similarity $p$ (frequency-based), and (ii) the similarity of the

---

[5]Our analysis does not contrast structural similarity with surface similarity, since it operates across representations. Instead, it contrasts the number of structurally related learned items with the

strength of the top structurally aligned examples.

[6]In practice, our algorithm can be described by: $AS(G_b, G_t) = 0.4\left(\frac{||(V_b, w_b) \cap (V_t, w_t)||}{||(V_b, w_b) \cup (V_t, w_t)||}\right) + 0.6\left(\frac{|E_t \cap E_b|}{|E_t \cup E_b|}\right)$.

*Table 3.* Highest supports for frequency- ($\rho_p$) and analogy-based ($\rho_k$) hypotheses for Qwen2.5-1.5B-Instruct on different training and test sets. All results are statistically significant. Analogy-based hypothesis outperforms the frequency-based one in successful generalizations.

| Train | NL | | Graph | | Graph→NL | |
|---|---|---|---|---|---|---|
| Test | NL | Graph | NL | Graph | NL | Graph |
| Frequency-based ($\rho_p$) | 0.176 | 0.124 | - | 0.188 | 0.245 | 0.273 |
| Analogy-based ($\rho_k$) | 0.242 | 0.148 | - | 0.291 | 0.265 | 0.297 |

$k$-th most similar item in the successfully learned training instances (analogy-based). We sweep $k \in \{1, ..., 10\}$ and $p \in \{0.1, ..., 0.9\}$ and use Pearson's $\rho$ as the correlation method. For each hypothesis, we report the statistically significant results with the highest support in Table 3, and full results in Appendix A.8.

Across all settings, we find that the analogy-based hypothesis consistently shows stronger correlations with test performance than the frequency-based one.[7] In particular, analogy explains the transfer better than frequency ($\rho_k$ 0.148 vs. $\rho_p$ 0.124) for cross-representation generalization from NL to Graph. When trained on Graph and testing on NL, neither hypothesis explains the results, consistent with the lack of visible transfer in this setting. Our curriculum further amplifies the analogical behavior ($\rho_k$ 0.265 vs. $\rho_p$ 0.245). Moreover, NL test performance after curriculum training is more correlated with Graph within-representation training than NL within-representation training ($\rho = 0.526$ vs. 0.353, both with p-value $<0.001$).[8] This suggests that symbolic warm-up specifically encourages the learning of procedures that later transfer across representations.

### 5.2. Qualitative Analysis

We qualitatively examine NL instances of Qwen2.5-1.5B-Instruct after NL, Graph, and Graph→NL training. We find that when trained on Graph only, the model does not recognize the underlying graph structure in NL, and simply defaults to summing up all the time constraints (Example A.7.1). When trained with NL data, the model understands that it has to search for the critical path (e.g., explicitly spelling out *Let's calculate the total time for this critical path*). Comparing instances that are wrong when trained on NL-only but correct when trained on the curriculum, the model only picks one path in the vanilla NL training, whilst

in the curriculum it learns to iterate across multiple possible paths and compares them to find the longest path (Example A.7.2). This finding indicates that symbolic induction encourages more explicit and systematic reasoning (Tversky & Kahneman, 1974; Xiang et al., 2025), rather than using a "lazy reasoning regime" when faced with very long (or perhaps high complexity) inputs (La Malfa et al., 2024). Even so, Graph→NL still suffers from errors such as wrong time unit conversion (Example A.7.3), indicating a performance ceiling caused by the base model capability cap.

In sum, our analysis demonstrates that successful cross-representation generalization is better predicted by structural similarity than frequency effects, hence can be interpreted as a form of generative analogy. Our curriculum improves performance by effectively encouraging it. We further compare between analogical and easy-to-hard (Hase et al., 2024) generalization in Appendix A.9, showing that while easy-to-hard generalization can benefit from within-representation easy data alone, cross-representation analogical generalization is better facilitated with a mixed data construction.

## 6. Related Works

**Learning procedures.** One core aspect of generalizing reasoning is to learn widely applicable procedures (Mitchell et al., 1986; Lewis, 1988), which requires meta-reasoning (i.e., reasoning about reasoning) (Russell & Wefald, 1991; Ackerman & Thompson, 2017; Griffiths et al., 2019). Procedural knowledge is influential for reasoning questions during pre-training (Ruis et al., 2024), and is important for the robust generalization of machine learning systems (Lin et al., 2024b). Our work builds on previous works to study post-training-time procedural transfer in a carefully controlled setting by varying data surface forms, while preserving the underlying procedural structures.

**Analogical reasoning of LLMs.** Analogical reasoning of LLMs has been primarily studied on proof-of-concept tasks such as textual/visual sequence manipulation (Mirchandani et al., 2023; Lewis & Mitchell, 2024; Musker et al., 2025) or by retrieving task-relevant examples via prompting (Yasunaga et al., 2023; Yu et al., 2023) with exemplar-based in-context learning (Shepard & Chang, 1963). The main focus tends to be on *proportional analogy* with simple relations and high surface similarity (Mikolov et al., 2013; Petersen &

---

[7]Another theoretical framework to consider is kernel-based generalization, which shares a similar spirit with analogy-based generalization if we interpret structural similarity as implicit kernels. LLMs' learning can be considered as bridging procedures across different representations, such as learning a feature map that induces a kernel to put similar procedures together.

[8]While the effect sizes are moderate, this is expected since we are not considering many other sources of variance (e.g., surface form, model stochasticity, and representation complexity), and our analysis isolates the underlying procedure as a single explanatory factor.

van der Plas, 2023; Yang et al., 2025). In contrast, *generative analogy* which involves richer relational mapping beyond surface similarity (Gentner, 1981; 1983; David Wong, 1993; Holyoak & Thagard, 1996) has received relatively less attention (see discussions in Yuan et al. (2023a), Hofmann et al. (2024), and Sultan et al. (2024)). In this work, we study generative analogy with rich stimuli via parametric learning and show that cross-representation generalization in LLMs can be interpreted as a form of generative analogy.

**Symbolic induction.** Training on code is generally helpful during LLM pre-training (Muennighoff et al., 2023; Aryabumi et al., 2024; Kim et al., 2024) and post-training for symbolic reasoning (MA et al., 2024; Zhang et al., 2025) and agentic planning (Chen et al., 2024). However, code training can sometimes hurt model performance in natural language tasks (Petty et al., 2024; Kotha et al., 2024). Similarly, works on graph training show mixed results (Zhang et al., 2024; Ye et al., 2024). Existing observations tend to be at the task level, offering limited insights into when and why learning on symbolic representations (e.g., code/graph) helps natural language problem solving. We study this question with isomorphic data design, and reveal that effective generalization occurs when LLMs can analogically learn from highly similar training instances.

**Generalization of RL and SFT in LLMs.** SFT and RL are commonly used in post-training of LLMs to acquire general capabilities (Ouyang et al., 2022; Wei et al., 2022a; Touvron et al., 2023). While some studies find that careful data selection can enable SFT to have superior performance (Ye et al., 2025), others report that RL generalizes better in out-of-domain settings (e.g., with varied game rules; Chu et al., 2025). Here, we compare SFT and RL methods in cross-representation generalization to explore whether and how these paradigms affect procedural generalization.

## 7. Conclusion

In this paper, we examine whether LLMs can learn generalizable procedures across representations in code, graphs, and natural language. Using isomorphic data to isolate procedure transfer from surface form, we show that learning symbolic data alone does not reliably transfer procedures to natural language problem solving. We propose a two-stage RL curriculum that trains first on symbolic and then natural language data. It substantially improves generalization across models and tasks. Last, we find that the successful cross-representation generalization can be interpreted as a form of generative analogy, which our curriculum effectively encourages. In all, these findings suggest that although LLMs can generalize across representations with appropriate curricula, they have to do so via extended training. This differs markedly from human analogical rea-

soning, which can show cross-representation generalization with only minimal exposure.

## Impact Statement

This paper presents work whose goal is to advance the field of Machine Learning. The societal consequences of our work include that it reveals critical learning mechanisms of LLMs in terms of generalizing structural data, which is especially interesting at a time when LLMs are heavily trained and tested in symbolic domains such as code and graphs. Also, we consider both the main `NL` dataset and its dialect variant `NL-AAVE` for technological fairness. However, we do observe there is still a notable gap between `NL` and `NL-AAVE` performance. This indicates that better solutions can be proposed to mitigate technological unfairness. Next, although we find that training with one or two representations does not generalize well to new representations, it remains an open question whether scaling representations at training time can bring an observable advantage. Last, we also note that LLMs still fall short of cross-representation generalization: while humans can often generalize in few- or even zero-shot, LLMs still need extensive in-representation training.

## Data Access Statement

The dataset used in this paper can be found here.

## Acknowledgment

We thank all the bodies who have provided funding for the authors and for the associated project. FL is supported by a Clarendon studentship. ZD is supported by the DARPA program SciFy. AGC is supported by the Fundamental Research priority area of The Alan Turing Institute, by the Special Funds of Tongji University for the "Sino-German Cooperation 2.0 Strategy", and by the EPSRC under grant EP/Z003512/1. Last, we are grateful to the people who offered invaluable feedback and suggestions along the way, and in particular to all reviewers of this paper.

The project on which this publication is based was funded by the Federal Ministry of Research, Technology and Space under the funding code "KI-Servicezentrum Berlin-Brandenburg" 16IS22092. Responsibility for the content of this publication remains with the author.

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

# A. Appendix

## A.1. Prompt example

NL

**Prompt**  To Make Salata Balati, here are the steps and the times needed for each step. Step 1. Prepare your Salad. (30 min) Step 2. Prepare the dressing. (10 min) Step 3. Dress the salad & serve. (5 min)

These ordering constraints need to be obeyed when executing above steps: Step 1 must precede step 3. Step 2 must precede step 3.

Question: Assume that you need to execute all the steps to complete the task and that infinite resources are available. What is the shortest possible time to Make Salata Balati? Think step by step. Then, encode your final answer in `<answer></answer>` (e.g. `<answer>1 min</answer>`)

**Answer**  35 min

NL-AAVE

**Prompt**  Say you wanna whoop up some Salata Balati, here's what you got to do and the times needed for each step. Step 1. whoop up your Salad. (30 min) Step 2. get that salad dressing together. (10 min) Step 3. Dress the salad and fix a plate for yourself. (5 min)

These ordering constraints gotta be followed when you doin' 'em steps above: You gotta deal with 1 before hittin' the 3. You gotta deal with 2 before hittin' the 3.

Question: Assumin' you outta do all 'em steps to finish up the task, and you got infinite resources. What the shortest time be to knock this task out? Aight, let's break it down step by step. Then wrap that answer up in `<answer></answer>` (e.g., `<answer>1 min</answer>`).

**Answer**  35 min

Graph

**Prompt**  You have a graph whose adjacency list representation is as follows:

{'1': ['3'], '2': ['3'], '3': ['END'], 'END': [], 'START': ['1', '2']}

The graph is a directed graph, and the nodes are labelled as follows (START and END are special nodes which takes no time to traverse): {'1': '30 min', '2': '10 min', '3': '5 min'}

Suppose you have to traverse from node 'START' to node 'END', how long does the longest path take? Think step by step. Then, encode your final answer in `<answer></answer>` (e.g. `<answer>1 min</answer>`)

**Answer**  35 min

Code

**Prompt**  Below is a Python function to search for the longest path from a source node to a target node in a directed acyclic graph (DAG) using the adjacency list representation.

The function takes a weighted adjacency list (a dictionary mapping each source node $i$ to a list of $(j, w)$ pairs, where $j$ is a target node and $w$ is the weight of the edge), along with a source and target node, and returns the longest path length from source to target.

```
import networkx as nx

def find_longest_path_from_source_to_target(weighted_adj_list,
source, target):
```

```
G = nx.DiGraph()
for src, neighbors in weighted_adj_list.items():
    for tgt, weight in neighbors:
        G.add_edge(src, tgt, weight=weight)

topo_order = list(nx.topological_sort(G))
dist = {node: float('-inf') for node in G.nodes}
pred = {node: None for node in G.nodes}
dist[source] = 0

for u in topo_order:
    for v in G.successors(u):
        weight = G[u][v]['weight']
        if dist[u] + weight > dist[v]:
            dist[v] = dist[u] + weight
            pred[v] = u

if dist[target] == float('-inf'):
    return None, []

path = []
current = target
while current is not None:
    path.append(current)
    current = pred[current]
path.reverse()

return dist[target]
```

Suppose your inputs are as follows:

```
adj_list = {
    639: [(621, 5.0)],
    339: [(621, 5.0)],
    621: [(833, 0.0)],
    833: [],
    811: [(639, 30.0), (339, 10.0)]
}
source = 811
target = 833
```

Think step by step. Then, encode the output of the function in <answer></answer> (e.g., <answer>1</answer>).

**Answer**  35.0

## A.2. Additional Results on More Model Families and Tasks

### A.2.1. LLAMA-3/OLMO-2 RESULTS

We train Llama-3.2-1/3B, Llama-3.1-8B, and Olmo-2-1/7B with the same experiment setting as we report in Section 3. We find that Llama-3 and Olmo-2 in general fail to generalize across representations when trained on symbolic tasks only, in line with our findings in the main content.

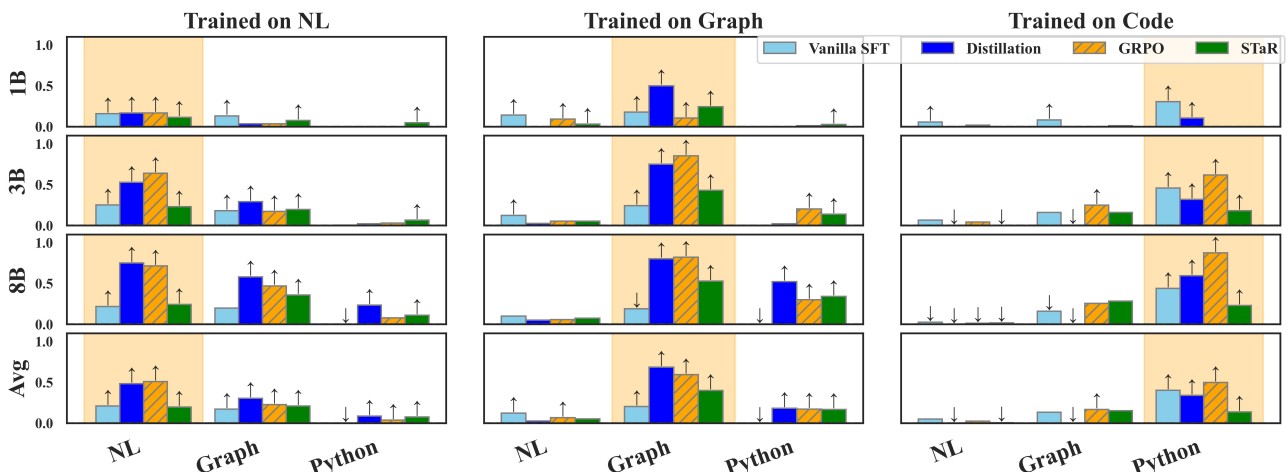

*Figure 3.* Main results for Llama models after training. We train models on one training set from `NL`, `Graph`, and `Code`, respectively (left to right), and test them on all settings (with the particular aim to optimize on `NL` representations). Results in **orange** background are in the same distribution as the training set (e.g., train on `NL` and test on `NL`). Up/down arrows denote test results significantly better/worse than untuned baselines by McNemar's tests (McNemar, 1947). Llama-1B cannot sample meaningful results for STaR method on `Code`, so we report the baseline result for the corresponding space. In general, we continue to witness our findings in the main content that there is no meaningful transfer when trained on a single representation.

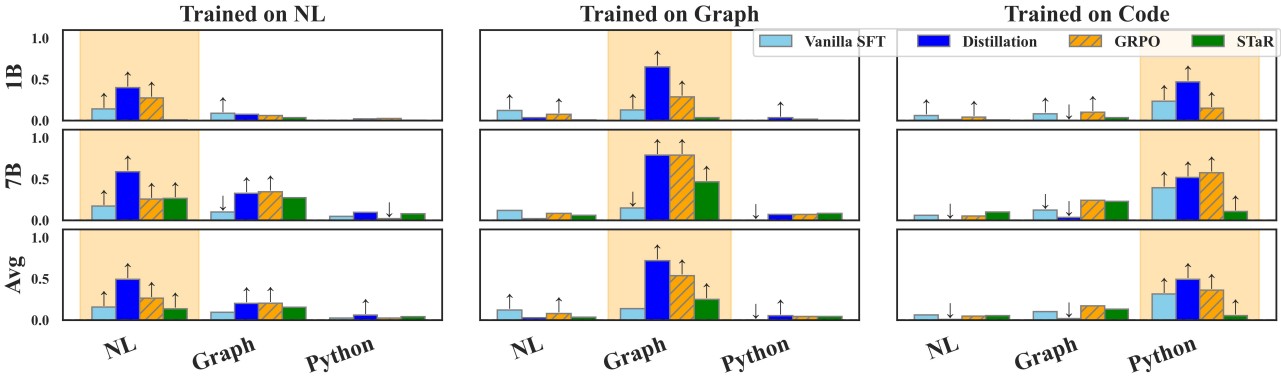

*Figure 4.* Main results for Olmo models after training. We train models on one training set from `NL`, `Graph`, and `Code`, respectively (left to right), and test them on all settings (with the particular aim to optimize on `NL` representations). Results in **orange** background are in the same distribution as the training set (e.g., train on `NL` and test on `NL`). Up/down arrows denote test results significantly better/worse than untuned baselines by McNemar's tests (McNemar, 1947). Olmo-1B cannot sample meaningful results for STaR method, so we report the baseline result for the corresponding space. In general, we continue to witness our findings in the main content that there is no meaningful transfer when trained on a single representation.

### A.2.2. MATH/PHYSICS RESULTS

We additionally train Qwen-2.5-1.5B-Instruct on math and physics data in `NL` and `Code` in Huang et al. (2025) for 20 steps each setting. As reported in Table 4, we observe again that training on `Code` alone does not naturally transfer to `NL`.

### A.3. Implementation Details

All experiments are run on 4×80GB H100. Input max length is 2048, and max length of new generated tokens is 6000 (except Olmo whose cutoff length is 4096). GRPO experiments are run with a train batch size 128 and a rollout batch size 512, initial kl coefficient is 0.01 (we also tested with 0.05 and observed no significantly different result trends). Top-p, gamma, and lambda are set to 1.0. Evaluations are done with temperature=0 and only one response for each prompt.

*Table 4.* Results of testing Qwen-1.5B-Instruct trained on one representation among Physics-`NL`/`Code`/Math-`NL`/`Code` and all other settings. Best results in each test setting are marked **bold**. We observe again that training on a symbolic representation does not naturally transfer to other representations.

| Train | | Test | | | |
|---|---|---|---|---|---|
| | | Physics | | Math | |
| | | NL | Code | NL | Code |
| Physics | NL | **0.475** | 0.490 | 0.275 | 0.315 |
| | Code | 0.280 | **0.595** | 0.310 | 0.415 |
| Math | NL | 0.155 | 0.460 | **0.405** | 0.485 |
| | Code | 0.180 | 0.425 | 0.240 | **0.585** |

### A.3.1. DISTILLATION IMPLEMENTATION DETAILS

We experiment with two teacher models in distillation: QwQ-32B (Team, 2025; Yang et al., 2024), DeepSeek-R1-Distill-Qwen-32B (DeepSeek-AI, 2025). We use temperature = 1, max_gen_len = 6000, and initial n_samples = 4 and experiment with both models to see if we can sample at least one correct output for each instance in our training set. We find that DeepSeek-R1-Distill-Qwen-32B largely outperforms QwQ-32B in sampling answers in this stage (0.919 vs. 0.793 for Pass@4), and is also less likely to output repeated sequences in the reasoning chain. We therefore choose DeepSeek-R1-Distill-Qwen-32B as the teacher model, and use 5-shot prompting in the later stages to iteratively generate more outputs for instances without training data (n_samples = 4 , 8, 16, 32, 64) for 5 rounds. We randomly sample one from the outputs for each instance with correct answers, and omit instances that do not have a correct answer to formulate a fine-tuning dataset. Finally, we obtain a distillation dataset of 1363 data points (out of 1373 original data points in total).

### A.4. Reward Curve

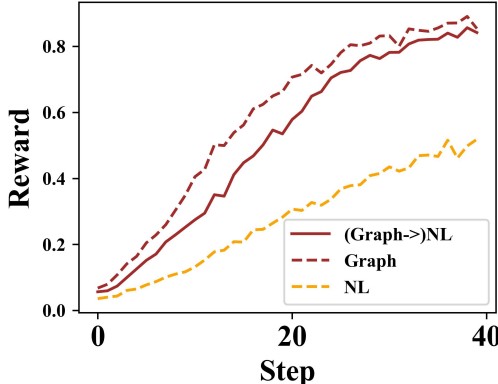

*Figure 5.* Reward curve for `NL`, `Graph` training from cold start, and `(Graph->)NL` which is `NL` training initialized with the final checkpoint of `Graph`. The learning curve of `(Graph->)NL` resembles `Graph` more than `NL`.

## A.5. Main Performance Delta Results

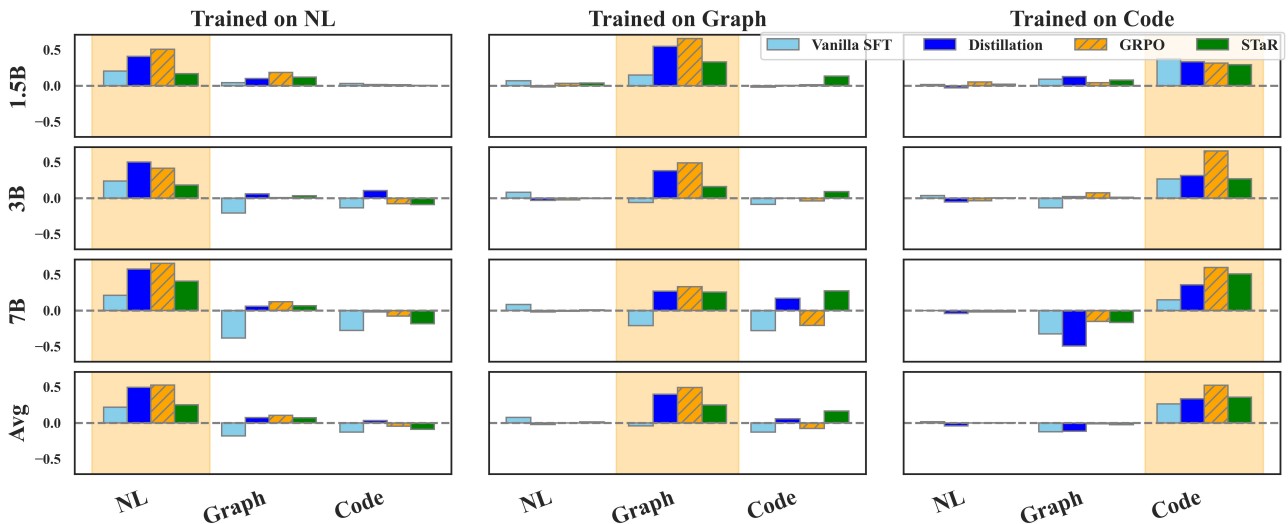

*Figure 6.* Qwen performance change (before and after training) for the main experiments. We train models on one representation from `NL`, `Graph`, and `Code`, respectively (left to right), and test them on all representations (with the particular aim to optimize on `NL`). Results in **orange** background are in the same distribution as the training set (e.g., train on `NL` and test on `NL`). Distillation can sometimes exacerbate performance degradation across representations (e.g., training on `Code` and testing on `Graph` for Qwen-2.5-7B-Instruct.)

## A.6. Curriculum on Olmo-2-7B-Instruct

*Table 5.* Test accuracy on `NL` and `NL-AAVE` datasets for different training strategies on Olmo-2-7B-Instruct. Best results are in **bold**. Results with $^*$ are statistically significant improvement over 80-step `NL` by McNemar's tests (McNemar, 1947). Our method outperforms the baseline method to train on `NL` only, both in the training representation and in the dialect variation.

| Train Setting | Test Acc. (NL) | Test Acc. (NL-AAVE) |
|---|---|---|
| Graph (40 steps) + NL (40 steps) | **0.631** | **0.569**$^*$ |
| NL only (80 steps) | 0.613 | 0.431 |

## A.7. Qualitative Examples

### A.7.1. EXAMPLE 1: NAIVELY ADDING NUMBERS

**Prompt**

To try fishing for the first time, here are the steps and the times needed for each step.
Step 1. Drive to the outdoor store (10 minutes)
Step 2. Compare fishing poles (30 minutes)
Step 3. Buy a fishing pole (5 minutes)
Step 4. Buy some bait (5 minutes)
Step 5. Drive to a lake (20 minutes)
Step 6. Rent a small boat (15 minutes)

These ordering constraints need to be obeyed when executing the above steps:
Step 1 must precede Step 2.
Step 2 must precede Steps 3 and 4.
Step 3 must precede Step 5.
Step 4 must precede Step 5.
Step 5 must precede Step 6.

**Question:** Assume that you need to execute all the steps to complete the task and that infinite resources are available. What is the shortest possible time to try fishing for the first time? Think step by step. Then, encode your final answer in `<answer></answer>` (e.g., `<answer>1 min</answer>`).

To determine the shortest possible time to try fishing for the first time, we need to consider the dependencies between the steps and find the sequence that minimizes the total time.

**Steps and durations:**
Step 1: Drive to the outdoor store (10 minutes)
Step 2: Compare fishing poles (30 minutes)
Step 3: Buy a fishing pole (5 minutes)
Step 4: Buy some bait (5 minutes)
Step 5: Drive to a lake (20 minutes)
Step 6: Rent a small boat (15 minutes)

**Dependencies:**
Step 1 must precede Step 2.
Step 2 must precede Steps 3 and 4.
Step 3 must precede Step 5.
Step 4 must precede Step 5.
Step 5 must precede Step 6.

**Execution sequence:**
Step 1: 10 minutes
Step 2: 30 minutes
Step 3: 5 minutes
Step 4: 5 minutes
Step 5: 20 minutes
Step 6: 15 minutes

**Total time:** $10 + 30 + 5 + 5 + 20 + 15 = 85$ minutes

`<answer>85 minutes</answer>`

To buy some headphones, here are the steps and the times needed for each step.

Step 1: Find out what headphones are out there (15 minutes)
Step 2: Look at the headphone reviews (15 minutes)
Step 3: Watch a video review (10 minutes)
Step 4: Add the headphones to cart (1 minutes)
Step 5: Enter in credit card details (3 minutes)

These ordering constraints need to be obeyed when executing above steps:
Step 1 must precede Step 2 and 3.
Step 2 must precede Step 4.
Step 3 must precede Step 4.
Step 4 must precede Step 5.

Question: Assume that you need to execute all the steps to complete the task and that infinite resources are available. What is the shortest possible time to buy some headphones? Think step by step. Then, encode your final answer in `<answer></answer>` (e.g. `<answer>1 min</answer>`).

To determine the shortest possible time to buy some headphones, we need to consider the dependencies between the steps and find the most efficient order in which to execute them.

Step 1: Find out what headphones are out there (15 minutes)
Step 2: Look at the headphone reviews (15 minutes)
Step 3: Watch a video review (10 minutes)
Step 4: Add the headphones to cart (1 minute)
Step 5: Enter in credit card details (3 minutes)

Now, let's consider the dependencies:
Step 1 must precede Step 2 and Step 3.
Step 2 must precede Step 4.
Step 3 must precede Step 4.
Step 4 must precede Step 5.

To minimize the total time, we should execute the steps in the most efficient order. Let's break it down:

Step 1: Find out what headphones are out there (15 minutes)
Step 2: Look at the headphone reviews (15 minutes)
Step 3: Watch a video review (10 minutes)
Step 4: Add the headphones to cart (1 minute)
Step 5: Enter in credit card details (3 minutes)

The total time is the sum of the times for each step:
15 minutes + 15 minutes + 10 minutes + 1 minute + 3 minutes = 44 minutes

Therefore, the shortest possible time to buy some headphones is 44 minutes.

```
<answer>44 minutes</answer>
```

### A.7.2. EXAMPLE 2: LAZY REASONING IN NL VS. PATH ITERATION IN (GRAPH−>)NL

Prompt

To mow the lawn today, here are the steps and the times needed for each step.
Step 1. Get the lawnmower out of the garage (5 minutes)
Step 2. Make sure the mower has enough gas (5 minutes)
Step 3. Check that all the parts are still working (15 minutes)
Step 4. Start the mower (5 minutes)
Step 5. Mow the lawn (1 hour)

These ordering constraints need to be obeyed when executing the above steps:
Step 1 must precede Steps 2 and 3.
Step 2 must precede Step 4.
Step 3 must precede Step 4.
Step 4 must precede Step 5.

**Question:** Assume that you need to execute all the steps to complete the task and that infinite resources are available. What is the shortest possible time to mow the lawn today? Think step by step. Then, encode your final answer in `<answer></answer>` (e.g., `<answer>1 min</answer>`).

Answer (trained on NL)

To find the shortest possible time to mow the lawn, we need to consider the dependencies between the steps and determine the critical path. Let's break it down step by step.

Step 1: Get the lawnmower out of the garage (5 minutes)
Step 2: Make sure the mower has enough gas (5 minutes)
Step 3: Check that all the parts are still working (15 minutes)
Step 4: Start the mower (5 minutes)
Step 5: Mow the lawn (1 hour = 60 minutes)

Now, let's visualize the dependencies:
Step 1 must precede Step 2 and Step 3.
Step 2 must precede Step 4.
Step 3 must precede Step 4.
Step 4 must precede Step 5.

The critical path will be:
Step 1 → Step 2 → Step 4 → Step 5

Let's calculate the total time for this critical path:
Step 1: 5 minutes
Step 2: 5 minutes
Step 4: 5 minutes
Step 5: 60 minutes

Total time = 5 + 5 + 5 + 60 = 75 minutes

```
<answer>75 minutes</answer>
```

Answer (trained on `(Graph->)NL`)

To determine the shortest possible time to mow the lawn today, we need to consider the dependencies between the steps and find the critical path. Let's break down the steps and their dependencies.

Step 1: Get the lawnmower out of the garage (5 minutes)
Step 2: Make sure the mower has enough gas (5 minutes)
Step 3: Check that all the parts are still working (15 minutes)
Step 4: Start the mower (5 minutes)
Step 5: Mow the lawn (1 hour = 60 minutes)

Now, let's analyze the dependencies:
Step 1 must precede Step 2 and Step 3.
Step 2 must precede Step 4.
Step 3 must precede Step 4.
Step 4 must precede Step 5.

We need to find the longest path that follows these dependencies.

Path 1: Step 1 → Step 2 → Step 4 → Step 5
Time: 5 minutes + 5 minutes + 5 minutes + 60 minutes = 75 minutes

Path 2: Step 1 → Step 3 → Step 4 → Step 5
Time: 5 minutes + 15 minutes + 5 minutes + 60 minutes = 85 minutes

Comparing the two paths, the longest path is 85 minutes.

Therefore, the shortest possible time to mow the lawn today is 85 minutes.

```
<answer>85 minutes</answer>
```

A.7.3. EXAMPLE 3: WRONG TIME UNIT CONVERSION IN GRAPH→NL

Prompt

To collect bacteria samples, here are the steps and the times needed for each step.

Step 1: Wear suitable left glove (20 seconds)
Step 2: Wear suitable right glove (20 seconds)
Step 3: Cut a small piece of the suspect material (15 minutes)
Step 4: Place the sample inside a clean plastic bag (10 seconds)
Step 5: Close the plastic bag (5 seconds)
Step 6: Label the plastic bag (2 minutes)
Step 7: Store the sample (20 minutes)

These ordering constraints need to be obeyed when executing above steps:
Step 1 must precede Step 3.
Step 2 must precede Step 3.
Step 3 must precede Step 4.
Step 4 must precede Step 5.
Step 5 must precede Step 6.
Step 6 must precede Step 7.

Question: Assume that you need to execute all the steps to complete the task and that infinite resources are available. What is the shortest possible time to collect bacteria samples? Think step by step. Then, encode your final answer in `<answer></answer>` (e.g. `<answer>1 min</answer>`).

Answer (trained on GRAPH→NL)

To determine the shortest possible time to collect bacteria samples, we need to consider the dependencies between the steps and find the critical path. Let's break down the steps and their dependencies.

Step 1: Wear suitable left glove (20 seconds)
Step 2: Wear suitable right glove (20 seconds)
Step 3: Cut a small piece of the suspect material (15 minutes)
Step 4: Place the sample inside a clean plastic bag (10 seconds)
Step 5: Close the plastic bag (5 seconds)
Step 6: Label the plastic bag (2 minutes)
Step 7: Store the sample (20 minutes)

Now, let's analyze the dependencies:
Step 1 must precede Step 3.
Step 2 must precede Step 3.
Step 3 must precede Step 4.
Step 4 must precede Step 5.
Step 5 must precede Step 6.
Step 6 must precede Step 7.

We need to find the longest path that follows these dependencies.

Path 1: Step 1 → Step 3 → Step 4 → Step 5 → Step 6 → Step 7
Time: 20 seconds + 15 minutes + 10 seconds + 5 seconds + 2 minutes + 20 minutes = 20 seconds + 15 minutes + 10 seconds + 5 seconds + 2 minutes + 20 minutes = 15 minutes + 20 minutes + 2 minutes + 20 seconds + 10 seconds + 20 seconds = 55 minutes + 30 seconds = 55 min-

utes 30 seconds

Path 2: Step 2 → Step 3 → Step 4 → Step 5 → Step 6 → Step 7
Time: 20 seconds + 15 minutes + 10 seconds + 5 seconds + 2 minutes + 20 minutes = 20 seconds + 15 minutes + 10 seconds + 5 seconds + 2 minutes + 20 minutes = 15 minutes + 20 minutes + 2 minutes + 20 seconds + 10 seconds + 20 seconds = 55 minutes + 30 seconds = 55 minutes 30 seconds

Both paths have the same total time. Therefore, the shortest possible time to collect bacteria samples is 55 minutes 30 seconds.

```
<answer>55 minutes 30 seconds</answer>
```

## A.8. Details of Frequency- and Analogy-based Hypothesis Test Results

*Table 6.* Details of frequency- and analogy-based hypothesis test results.

| Train | Test | K | $\rho_k$ | p-value$_k$ | P | $\rho_p$ | p-value$_p$ |
|---|---|---|---|---|---|---|---|
| **Train: Graph** | **Graph** | 1 | 0.247 | <0.001 | 0.1 | -0.041 | 0.542 |
| | | 2 | 0.242 | <0.001 | 0.2 | -0.155 | 0.020 |
| | | 3 | 0.232 | <0.001 | 0.3 | -0.125 | 0.061 |
| | | 4 | 0.211 | 0.001 | 0.4 | **0.188** | 0.005 |
| | | 5 | 0.218 | <0.001 | 0.5 | 0.166 | 0.013 |
| | | 6 | 0.220 | <0.001 | 0.6 | 0.099 | 0.137 |
| | | 7 | 0.218 | <0.001 | 0.7 | 0.096 | 0.150 |
| | | 8 | 0.214 | 0.001 | 0.8 | 0.096 | 0.150 |
| | | 9 | 0.289 | <0.001 | 0.9 | 0.096 | 0.150 |
| | | 10 | **0.291** | <0.001 | | | |
| | **NL** | 1 | -0.047 | 0.486 | 0.1 | -0.048 | 0.477 |
| | | 2 | -0.071 | 0.286 | 0.2 | 0.001 | 0.994 |
| | | 3 | -0.065 | 0.332 | 0.3 | -0.070 | 0.298 |
| | | 4 | -0.071 | 0.291 | 0.4 | -0.093 | 0.163 |
| | | 5 | -0.068 | 0.312 | 0.5 | -0.064 | 0.336 |
| | | 6 | -0.059 | 0.375 | 0.6 | -0.044 | 0.510 |
| | | 7 | -0.053 | 0.430 | 0.7 | -0.041 | 0.539 |
| | | 8 | -0.057 | 0.398 | 0.8 | -0.041 | 0.539 |
| | | 9 | -0.063 | 0.350 | 0.9 | -0.041 | 0.539 |
| | | 10 | -0.057 | 0.397 | | | |
| **(Graph->)NL** | **Graph** | 1 | 0.242 | <0.001 | 0.1 | 0.046 | 0.490 |
| | | 2 | 0.236 | <0.001 | 0.2 | 0.034 | 0.612 |
| | | 3 | 0.234 | <0.001 | 0.3 | 0.157 | 0.018 |
| | | 4 | 0.216 | 0.001 | 0.4 | **0.273** | <0.001 |
| | | 5 | 0.221 | <0.001 | 0.5 | 0.162 | 0.015 |
| | | 6 | 0.219 | <0.001 | 0.6 | 0.105 | 0.115 |
| | | 7 | 0.217 | 0.001 | 0.7 | 0.102 | 0.129 |
| | | 8 | 0.219 | <0.001 | 0.8 | 0.102 | 0.129 |
| | | 9 | **0.297** | <0.001 | 0.9 | 0.102 | 0.129 |
| | | 10 | 0.293 | <0.001 | | | |
| | **NL** | 1 | 0.230 | <0.001 | 0.1 | 0.039 | 0.563 |
| | | 2 | 0.213 | 0.001 | 0.2 | -0.015 | 0.821 |
| | | 3 | 0.209 | 0.002 | 0.3 | 0.068 | 0.310 |
| | | 4 | 0.196 | 0.003 | 0.4 | **0.245** | <0.001 |
| | | 5 | 0.215 | 0.001 | 0.5 | 0.142 | 0.033 |
| | | 6 | 0.210 | 0.002 | 0.6 | 0.113 | 0.092 |
| | | 7 | 0.203 | 0.002 | 0.7 | 0.109 | 0.104 |
| | | 8 | 0.203 | 0.002 | 0.8 | 0.109 | 0.104 |
| | | 9 | **0.265** | <0.001 | 0.9 | 0.109 | 0.104 |
| | | 10 | 0.262 | <0.001 | | | |
| **Train: NL** | **Graph** | 1 | 0.115 | 0.084 | 0.1 | 0.009 | 0.891 |
| | | 2 | **0.148** | 0.027 | 0.2 | -0.089 | 0.184 |
| | | 3 | 0.147 | 0.027 | 0.3 | -0.041 | 0.543 |
| | | 4 | 0.104 | 0.119 | 0.4 | 0.112 | 0.094 |
| | | 5 | 0.048 | 0.476 | 0.5 | **0.124** | 0.064 |

**(Continued)**

| Train | Test | K | $\rho_k$ | p-value$_k$ | P | $\rho_p$ | p-value$_p$ |
|---|---|---|---|---|---|---|---|
| | | 6 | 0.117 | 0.079 | 0.6 | 0.018 | 0.789 |
| | | 7 | 0.118 | 0.077 | 0.7 | 0.039 | 0.563 |
| | | 8 | 0.118 | 0.078 | 0.8 | 0.039 | 0.563 |
| | | 9 | 0.100 | 0.136 | 0.9 | 0.039 | 0.563 |
| | | 10 | 0.104 | 0.120 | | | |
| | NL | 1 | 0.224 | <0.001 | 0.1 | -0.054 | 0.421 |
| | | 2 | 0.236 | <0.001 | 0.2 | -0.108 | 0.106 |
| | | 3 | **0.242** | <0.001 | 0.3 | -0.122 | 0.067 |
| | | 4 | 0.227 | <0.001 | 0.4 | 0.156 | 0.019 |
| | | 5 | 0.213 | 0.001 | 0.5 | 0.174 | 0.009 |
| | | 6 | 0.202 | 0.002 | 0.6 | **0.176** | 0.008 |
| | | 7 | 0.204 | 0.002 | 0.7 | 0.171 | 0.010 |
| | | 8 | 0.202 | 0.002 | 0.8 | 0.171 | 0.010 |
| | | 9 | 0.206 | 0.002 | 0.9 | 0.171 | 0.010 |
| | | 10 | 0.207 | 0.002 | | | |

## A.9. Comparison with Easy-to-hard Generalization

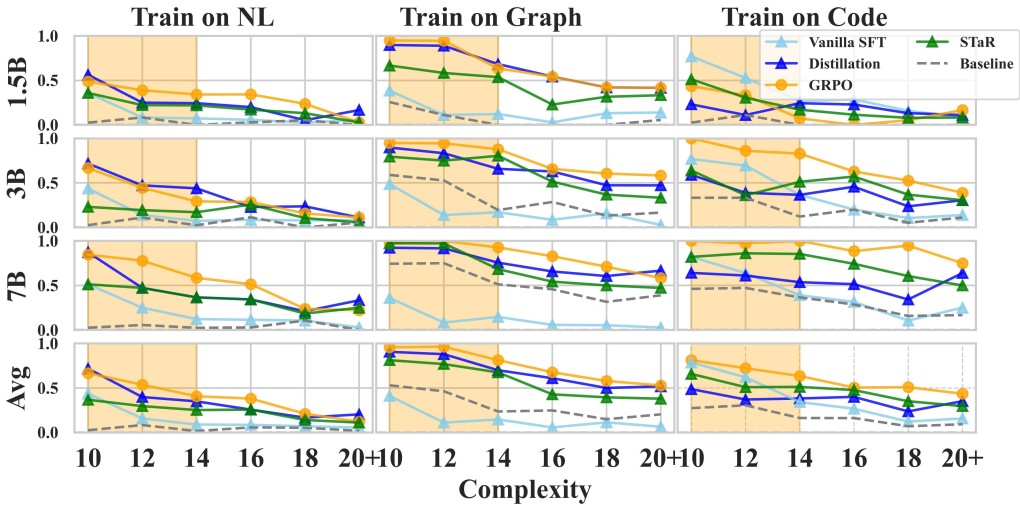

*Figure 7.* Test performance on the same representations of the training data after training of instances of complexity 14 or lower (here complexity means the total count of edges and nodes in each corresponding graph). Results are aggregated based on different complexity levels of the test cases. Results in **orange** background share the same range of complexity with training data.

We compare cross-representation generalization with *easy-to-hard generalization*. Conceptually, we emphasize that these two phenomena are orthogonal. Easy-to-hard generalization describes the same representation of procedures with different complexities (e.g., train on 3-step and test on 5-step planning, Figure 1 center and left) (Schwarzschild et al., 2021; Burns et al., 2024; Hase et al., 2024; Sun et al., 2024). Although synthesizing code (e.g., (Li et al., 2025)) can, in principle, overcome this problem by controlling the complexities of the programming problems, we see in Section 3 that it does not generalize across representations.

To empirically illustrate the difference, we train Qwen models on task instances of complexities 14 or lower according to the complexity measure in Section 2 (802 cases in total). We report test results by complexity in Figure 7. Method-wise, GRPO generalizes best across the board, similarly in cross-representation generalization. However, when looking at general patterns across methods, all of them provide a boost on hard instances even when only trained on easy ones (except vanilla SFT), in line with (Sun et al., 2024). All complexity levels show a decreasing trend from higher to lower complexities. This is in sharp contrast with our observation in cross-representation generalization, where we find almost zero or even negative

transfer across representations. Our observation indicates that cross-representation generalization is more challenging for current LLMs compared to in-representation easy-to-hard generalization. Despite so, training on our two-stage data curriculum enables significant knowledge transfer.

