# OpenReview forum: "Can Large Language Models Generalize Procedures Across Representations?"
_ICML.cc/2026/Conference — ICML 2026 regular_

### Official Review · Reviewer_YEtx · 2026-03-03

**Soundness:** 4
**Presentation:** 3
**Significance:** 3
**Originality:** 3
**Overall Recommendation:** 5
**Confidence:** 3

**Summary:**

The paper investigates whether (specific types of) procedural reasoning can be transfer learned from one representation, e.g. graphs or source code, and then applied in a different representation, e.g. natural language. The primary finding is that this transfer does not occur zero-shot across SFT, GRPO, and others, *but* that curriculum learning where training is first in e.g. code and then in e.g. NL does confer the desired transfer (interestingly, only with GRPO and not with SFT variants). They explain this as a kind of analogical generalization, and provide quantitative analyses suggesting that the transfer effect is driven by specific training examples with high-analogical-resemblance to specific test examples, and not by a broad-based frequency effect. The implications are that RL may offer a better path to analogical generalization, but that such generalization is not zero-shot, and requires a curriculum. Even controlling for amount of data, the curriculum outperforms simply training on the ultimate task modality, so for example it is better to curriculum learn code->NL than to just train on NL for the same amount of data. Experiments are primarily on asynchronous planning, with supplemental experiments confirming the effect on math->physics problem transfer.

**Compliance With Llm Reviewing Policy:**

Affirmed.

**Final Justification:**

The rebuttal was helpful in the sense that it was nice to get responses to my high level questions about the work; I already liked the paper and still do.

**Key Questions For Authors:**

You make the valid point that your results show that this LLM-analogy-reasoning isn't few-shot like it is in humans. But what if the LLM had a curriculum of isomorphic reasoning problems across $K$ modalities, for large $K$? You investigate $K=1$, but perhaps it could analogize zero-shot for large $K$.

**Limitations:**

yes

**Strengths And Weaknesses:**

Strengths: the paper is presented well and has clear implications for pretraining, and also suggests a mechanistic account of the observed effects. The fact that it is explained as a kind of analogy could help spur for their research in that area, which is relatively neglected. The experiments are comprehensive and confirmed the effect across a range of language models. Statistical analyses are thorough. Followup experiments in math->physics confirm that it is not a fluke of the centerpiece dataset. The scientific result is nonobvious: RL (they investigate GRPO) is necessary for the transfer effect suggesting that the language model might have to explore a round a it to discover how to link its earlier curriculum contents to the new problems. The paper also is refreshingly honest in its appraisal of the limitations in linking this to human-level analogical reasoning, saying "This stands in contrast with human analogical behaviors, which exhibit
cross-representation generalization with minimal exposure".

Weaknesses: They primarily investigate asynchronous planning, which is (I think?) kind of a weird nonstandard task compared to e.g. math problems. One weakness which is not *really* the author's fault is that many of these questions are *best* investigated at frontier scale. I sense from the paper that they are academics though and wouldn't be able to do that. I also do have one question described below that I hoped the others could discuss in their paper.

Nitpick: While I appreciate the explanation in terms of analogy, it would also be insightful to explain it as a kind of kernel-based generalization. There are many formal equivalences between neural networks and kernel machines.

---

> ### Author Rebuttal · Authors · 2026-03-30
>
> We thank the reviewer for acknowledging the clear fundamental impact of the paper, the novel analysis on LLM generalization patterns, the comprehensive experiment design, as well as the refreshing conclusions about the experiment results and relevant indications.
>
>
> To address your concerns:
>
>
> **Question**: *…what if the LLM had a curriculum of isomorphic reasoning problems across K  modalities, for large K? You investigate K=1, but perhaps it could analogize zero-shot for large K.*
>
>
> **Response**: This is an interesting question! We add new experiments to use Graph+Code as the training set, but we find that this setting does not make the model generalize zero-shot to NL either (acc 0.067). Furthermore, we experimented with using Graph+Code as the warm-up stage instead of Graph-only (keeping the second stage NL unchanged), but the result suggests that the Graph+Code→NL case is less strong compared to Graph→NL (acc 0.382 vs. 0.782 on NL), which we think is because the first stage fails to provide a strong enough symbolic base to learn from (0.522 in-representation acc for Graph+Code vs. 0.827 in-representation acc on Graph as first stage). However, there seems to be some evidence in the meta-learning space that sampling a large K is helpful at least in in-context learning [1], which makes it an open question regarding whether K exceeding a particular threshold would be helpful. We will make sure to add new experiment results and discussion in the final version of the paper, and we hope to see more follow-up work investigating this direction in the parametric learning space.
>
>
> **Weakness 1**: *They primarily investigate asynchronous planning, which is (I think?) kind of a weird nonstandard task compared to e.g. math problems.*
>
>
> **Response**: We thank the reviewer for asking about the choice for the main task. The reason why we are using the asynchronous planning task in our main experiment is that this task provides isomorphic constructions across three representations by construction, which is not readily available in other tasks, such as Math. Also, this task provides a realistic scenario about natural language queries by users, as the naturalistic data is sourced from WikiHow, and thus has a high degree of ecological validity. Please note that to provide more insights into other tasks, we also supplement with more results in Math and Physics in Section 4.3.
>
>
> **Weakness 2**: *…it would also be insightful to explain it as a kind of kernel-based generalization…*
>
>
> **Response**: This is a very interesting comparison! We agree that there is some similarity between what we argue in this paper and kernel-based generalization, especially if we interpret structural similarity as implicit kernels. We can imagine LLMs’ learning to bridge procedures across different representations as learning a feature map that induces a kernel to put similar procedures together. We will add relevant discussion in the final version of the paper!
>
>
> We deeply appreciate the reviewer’s efforts in giving feedback to help us improve the paper, and we are happy to discuss further!
>
>
> [1] Coda-Forno, J., Binz, M., Akata, Z., Botvinick, M., Wang, J., & Schulz, E. (2023). Meta-in-context learning in large language models. Advances in Neural Information Processing Systems, 36, 65189-65201.

---

> > ### Author Rebuttal · Reviewer_YEtx · 2026-04-03
> >
> > Thanks for the rebuttal, I continue to support this paper.

---

### Official Review · Reviewer_oWrW · 2026-03-09

**Soundness:** 3
**Presentation:** 2
**Significance:** 3
**Originality:** 3
**Overall Recommendation:** 4
**Confidence:** 3

**Summary:**

This paper asks whether LLMs can generalize procedural knowledge across different representations, namely Code, Graph, and natural language. To study this, the authors build an isomorphic setup where the underlying procedure is fixed and only the representation changes. The main finding is that training on symbolic representations alone does not reliably transfer to NL, even when within-representation performance is strong. To mitigate this, the paper proposes a simple two-stage training schedule: first tune on Graph to induce the underlying procedure, and then continue tuning on NL so that the learned procedure becomes usable in natural language. Experiments on planning, together with additional results on NL-AAVE, math, and physics, suggest that this setup works better than NL-only training. The paper also argues that the successful cases are better explained by generative analogy than by simple frequency-based learning.

**Compliance With Llm Reviewing Policy:**

Affirmed.

**Final Justification:**

The analysis is helpful, and some of the conclusions also give me additional thoughts; I suggest incorporating these discussions into the next version, and I have updated my score accordingly.

**Key Questions For Authors:**

- Would the authors agree that the contribution is more accurately described as improving target-format adaptation with symbolic warm-up, rather than achieving strong direct cross-representation transfer?
- Why was Graph chosen as the first-stage representation in the main experiments, instead of Code or a Graph+Code mixture? Were variants such as Code→NL, Graph+Code→NL, or mixed/interleaved training explored?
- If the final downstream target were Code rather than NL, would the same logic become Graph→Code or, more generally, symbolic→Code? If so, does that suggest that the main takeaway is that the model ultimately has to be adapted to the target representation?
- Could the authors provide more failure cases showing what NL-only, symbolic-only, and especially Graph→NL learn or fail to learn?
- The main two-stage gain appears to be demonstrated most clearly under full-parameter GRPO. Do the authors expect the same training strategy to work under standard SFT, prompt tuning, or other parameter-efficient adaptation methods?

Overall, I find the paper interesting, and I would be open to revising my assessment if the authors can convincingly address the above concerns in the rebuttal.

**Limitations:**

yes.

**Strengths And Weaknesses:**

Strengths
- I found the central question interesting. By holding the underlying procedure fixed and only changing the representation format, the paper isolates a useful and underexplored issue: symbolic training on Graph/Code does not seem to transfer to NL as naturally as one might expect.
- The proposed training strategy is simple, but the empirical improvement over the NL-only baseline is clear. Moreover, the paper does not stop at reporting the gain, and tries to give some explanation for why the two-stage setup helps.
- The evaluation is broader than a single planning benchmark. The authors also test NL-AAVE, math, and physics, and check similar trends on other model families such as Olmo, which makes the overall story more convincing.

Weaknesses
- My main reservation is about how the contribution is framed. The results seem to show that symbolic-only training does not directly transfer well to NL, and that an additional NL stage is still necessary. Because of this, the contribution feels closer to using symbolic warm-up to improve final NL adaptation than to demonstrating truly representation-agnostic procedural learning.
- I am also not fully convinced by the choice of Graph as the first-stage representation. The main story is built around Graph→NL, but it is not clear why Graph should be preferred over Code, or over a mixture of Graph and Code. Without these comparisons, it is hard to tell how much of the gain comes from the general idea and how much comes from this specific design choice.
- More broadly, the current evidence makes the method look somewhat target-format-dependent. Since the improvement only appears after continuing training on NL, one possible reading is simply that the model still needs explicit adaptation to the final target format. If that is the case, the scope of the claim should probably be stated more carefully.
- I would also have liked to see a bit more failure analysis. The paper gives some discussion, but it is still not very clear what NL-only, symbolic-only, and especially Graph→NL are actually learning differently, especially in the cases where they fail.
- Finally, I am unsure how general this training strategy is beyond the particular setting used here. Although the paper includes several SFT baselines, the main two-stage improvement seems most convincing under full-parameter GRPO, so I am not yet sure how broadly this carries over to other adaptation settings.

---

> ### Author Rebuttal · Authors · 2026-03-30
>
> Thank you for acknowledging the interest in the research question, the comprehensiveness of experiments, and the insight of our analysis.
>
>
> To address your questions:
>
> **Q1**:  *…the contribution is ...improving target-format adaptation with symbolic warm-up, rather than achieving strong direct cross-representation transfer?*
>
>
> **Response**: Thank you for asking for clarification. We agree that the last-stage training adapts LLMs and that LLMs do not directly generalize, and we will add the observation to the paper. However, this is not our sole contribution. Our contributions are three-fold: (i) demonstrating that procedures do not naively generalize, (ii) effective symbolic induction helps generalization, and (iii) interpreting successful generalization as analogy.
>
>
> **Q2**: *Why was Graph chosen as the first-stage…? Were variants explored?*
>
>
> **Response**: This is an interesting question! We perform new experiments to answer. As noted in the paper, the main inspiration for the curriculum is that some symbolic representations train much faster than NL (line 258-267): Graph (40 steps) vs. NL (40 steps) results in 0.827 vs. 0.538 acc in Qwen2.5-1.5B. However, Code does not show such an advantage (0.338 acc for 40 steps). Code (40 steps)→NL (40 steps) is even worse than NL-only (40 steps) (0.382 vs. 0.538 acc). Similarly, Graph+Code→NL is worse than Graph→NL (0.533 vs. 0.782 acc), because the first phase is too weak (0.522 acc). Last, interleaved Graph+NL (0.382 acc) is worse than NL-only. The insight is to start from an easy representation to establish a strong structural bias, then adapt. This is also in line with existing literature that curriculum outperforms joint training [2].
>
>
> **Q3**: *If the final downstream target were Code…, would the same logic become …symbolic→Code? Is the main takeaway task adaptation?*
>
>
> **Response**: Thank you for asking about downstream representations. Our new experiment suggests that Graph (40 steps)→Code (40 steps) (0.787 acc) is more effective than Code-only (80 steps) (0.373 acc) for Code. While adaptation is necessary, the symbolic warm-up choice is important (see Q2). Moreover, the adaptation is not our sole contribution (see Q1).
>
>
> **Q4**: *…more failure cases...?*
>
>
> **Response**: Thank you for asking for more qualitative cases.
>
>
> Graph→NL: the model learns systematic algorithmic patterns, but makes mistakes such as wrong time unit conversion:
>
>
> …Time: 20 seconds + 15 minutes + 10 seconds + 5 seconds + 2 minutes + 20 minutes
>
> …
>
> = 55 minutes + 30 seconds
>
> …
>
>
> NL-only:  the model can learn some dependencies, but it can naively choose a pattern without exhaustive comparisons, which we term ‘lazy reasoning’ (line 388-400):
>
>
> …- Step 1 must precede Step 2 and Step 3.
>
> - Step 2 must precede Step 4.
>
> - Step 3 must precede Step 4.
>
> …
>
>
> The critical path will be:
>
> Step 1 -> Step 2 -> Step 4 -> Step 5
>
>
> Graph-only: the model naively sums up the time for all steps, meaning that it does not understand the underlying NL dependencies (line 385-388):
>
> …
>
> - Step 1 must precede Step 2 and Step 3.
>
> - Step 2 must precede Step 4.
>
> - Step 3 must precede Step 4.
>
> …
>
>
> …The total time is the sum of the times for each step: 15 minutes + 15 minutes + 10 minutes + 1 minute + 3 minutes = 44 minutes
>
>
>
>
> **Q5**:  *… Do the authors expect the same training strategy to work under standard SFT or other PEFT methods?*
>
>
> **Response**: GRPO is best suited for the proposed curriculum. Changing second-stage GRPO to distillation significantly hurts model performance (0.782 vs. 0.462 acc, in paper lines 292 to 297).
>
>
> Similarly, this curriculum is not effective in standard SFT: (0.236 acc on NL-only, 2 epochs vs. 0.244 acc on NL-only, 4 epochs vs. 0.249 on Graph 2 epochs→NL 2 epochs). PEFT does not adapt well: changing the second-phase GRPO from full parameter to rank-8/16 LoRA makes Qwen2.5-1.5B perform with near-zero accuracy on NL. It suggests that successful structural transfer requires adequate computation and sufficient exploration through RL, in line with existing literature [1,3].
>
>
> We deeply appreciate the reviewer’s efforts in giving feedback to help us, and we will add relevant new results and discussions to the final paper. We hope that you could reconsider your score if our response addresses your concerns, and we are happy to discuss further!
>
>
>
>
> [1] Chu, T., Zhai, Y., Yang, J., Tong, S., Xie, S., Schuurmans, D., ... & Ma, Y. SFT Memorizes, RL Generalizes: A Comparative Study of Foundation Model Post-training. (2025). In Forty-second International Conference on Machine Learning.
>
>
> [2] Pentina, A., Sharmanska, V., & Lampert, C. H. (2015). Curriculum learning of multiple tasks. In Proceedings of the IEEE conference on computer vision and pattern recognition (pp. 5492-5500).
>
>
> [3] Shuttleworth, R. S., Andreas, J., Torralba, A., & Sharma, P. (2025). LoRA vs Full Fine-tuning: An Illusion of Equivalence. In The Thirty-ninth Annual Conference on Neural Information Processing Systems.

---

> > ### Author Rebuttal · Reviewer_oWrW · 2026-04-02
> >
> > Thanks, the analysis is helpful, and some of the conclusions also give me additional thoughts. I suggest incorporating these discussions into the next version, and I have updated my score accordingly.

---

### Official Review · Reviewer_3YqJ · 2026-03-12

**Soundness:** 2
**Presentation:** 3
**Significance:** 2
**Originality:** 2
**Overall Recommendation:** 4
**Confidence:** 3

**Summary:**

This paper studies whether post-trained LLMs can transfer procedures across natural language, graph, and code representations by converting asynchronous planning into isomorphic NL, graph, and code forms. It evaluates Qwen, Llama, and OLMo with vanilla SFT, distillation, STaR, and GRPO, and finds that symbolic-only training gives strong in-domain gains but weak transfer to NL. It then proposes a Graph→NL GRPO curriculum that improves Qwen-2.5-1.5B from 0.698 to 0.782 on NL, while also improving NL-AAVE from 0.507 to 0.573. The controlled setup is useful and the negative result is valuable.

**Compliance With Llm Reviewing Policy:**

Affirmed.

**Final Justification:**

The authors did a good job in addressing my concern, especially with adding extra experiments. As my major concern is addressed, I've updated my score.

**Key Questions For Authors:**

See weaknesses (especially W2).

**Limitations:**

See weaknesses. I would appreciate if more tasks beyond AsyncHow are compared.

**Strengths And Weaknesses:**

### Strengths

S1 The isomorphic NL, Graph, and Code construction is clean and insightful. It isolates representation shift while keeping the underlying DAG procedure fixed. This is the right experimental design for the stated question.

S2 The negative result is important. The paper shows that symbolic post-training does not automatically transfer to natural language, even when the underlying algorithm is unchanged. This aligns with prior findings that graph tuning often fails to generalize beyond in-distribution patterns, and that code data has mixed effects outside programming tasks.

S3 The curriculum result is genuinely interesting. On the main planning benchmark, Graph→NL improves the 1.5B Qwen model over NL-only training under the same step budget, and the improvement also carries to the dialect variant NL-AAVE.

S4 The qualitative examples are useful. They make the failure mode concrete. The graph-only model naively sums durations, while the curriculum model explicitly enumerates alternative paths and picks the true critical path.


### Weaknesses

W1  The task domain is narrow. The main benchmark reduces to longest path computation in a DAG (limited to AsyncHow). While this controlled setting is useful for analysis, it limits the strength of the claim about general procedure transfer across representations.

W2 The experimental comparison is limited. The curriculum result compares Graph→NL with NL only and NL→Graph, but does not include additional baselines such as mixed Graph and NL training or token matched training budgets. These controls would help determine whether the gains come from curriculum order or simply different exposure distributions.

W3 The mechanism claim about analogy based learning is not strongly supported. The analysis relies on a hand designed similarity metric derived from the same DAG structure that defines the task. The reported correlations are modest, so the evidence is suggestive rather than conclusive.

---

> ### Author Rebuttal · Authors · 2026-03-30
>
> We appreciate that the reviewer acknowledges the systematic design of our experiment and the comprehensive insights brought by our results and analysis.
>
>
> To address your concerns:
>
> **W1**: *The task domain is narrow… it limits the strength of the claim about general procedure transfer across representations.*
>
>
> **Response**: We thank the reviewer for asking about more task comparisons. Please note that while we focus on asynchronous planning in the main experiment for presentation clarity, we already supply additional results in Section 4.3 and Table 2 on page 6, where we show that our conclusions generalize to two additional domains, specifically Math and Physics. We will highlight these experiments better in the final version.
>
>
> **W2**: *The experimental comparison is limited. The curriculum result compares Graph→NL with NL only and NL→Graph, but does not include additional baselines such as mixed Graph and NL training or token matched training budgets…*
>
>
> **Response**: We thank the reviewer for requesting more baseline comparisons.
>
>
> For the concern about ‘additional baselines such as mixed Graph and NL training’, we perform additional experiments on interleaved Graph+NL training (i.e., using Graph+NL training data together for 40 gradient steps). While we control the same training budget, we find that the interleaved training underperforms Graph→NL curriculum (acc 0.382 vs. 0.782 on NL). This observation is in line with existing literature, which shows that learning multiple tasks at the same time can result in task interference and that curriculum learning can outperform joint training [2,3]. We will add relevant discussion to the final version of the paper.
>
>
> For the concern about ‘ token matched training budgets’, we note that in our training setting we fix the total training steps and sequence length cut, which means that essentially each condition is controlled with the same training token budget already. We will clarify this in the final version.
>
>
> **W3**: *The mechanism claim about analogy based learning is not strongly supported. The analysis relies on a hand designed similarity metric derived from the same DAG structure that defines the task. The reported correlations are modest, so the evidence is suggestive rather than conclusive.*
>
>
> **Response**: We appreciate the reviewer’s concern regarding the strength of analogy-based correlation. Our use of a DAG-derived similarity metric is intentional, as the DAG structure serves as the common construction basis of the dataset. The procedures in Code, Graph, and Natural Language share the same underlying DAG, making them structurally isomorphic and directly comparable at the level of underlying procedure. This measure is theoretically well supported by the principles of structure mapping [1], as we present in the paper Section 5.1 (line 349-361).
>
>
> Also, please note that we do not claim that analogical similarity fully explains model behavior, but rather that it provides a stronger account than frequency-based explanations (Section 5.1 in the main paper). This relative claim is directly supported by our experimental results, which consistently show stronger alignment with the analogy-based metric than with frequency baselines (Table 3 in the main paper). While it is true that the effect size is only moderate, this is expected since we are not considering many other sources of variance (e.g., surface form, model stochasticity, and representation complexity), and our analysis isolates the underlying procedure as a single explanatory factor. We will highlight this better in the final version of the paper.
>
>
> We deeply appreciate the reviewer’s efforts in giving feedback to help us improve the paper. We hope that you could reconsider your score if our response addresses your concerns, and we are happy to discuss further!
>
>
> [1] Gentner, D. (1983). Structure-mapping: A theoretical framework for analogy. Cognitive science, 7(2), 155-170.
>
>
> [2] Pentina, A., Sharmanska, V., & Lampert, C. H. (2015). Curriculum learning of multiple tasks. In Proceedings of the IEEE conference on computer vision and pattern recognition (pp. 5492-5500).
>
>
> [3] Standley, T., Zamir, A., Chen, D., Guibas, L., Malik, J., & Savarese, S. (2020, November). Which tasks should be learned together in multi-task learning?. In International conference on machine learning (pp. 9120-9132). PMLR.

---

> > ### Author Rebuttal · Reviewer_3YqJ · 2026-04-01
> >
> > Thank you for the detailed rebuttal and for providing additional evidence. I appreciate the effort to include further results on the extra interleaved Graph+NL comparison. These clarifications are helpful, and my main concern is now mostly addressed. Overall the rebuttal has improved my understanding of the paper. I've adjusted my score accordingly.

---

### Official Review · Reviewer_eL9F · 2026-03-13

**Soundness:** 3
**Presentation:** 3
**Significance:** 3
**Originality:** 4
**Overall Recommendation:** 5
**Confidence:** 4

**Summary:**

This paper investigates whether LLMs can generalize procedural knowledge across different representations of the same underlying "logic", probed by a task given as natural language, graph and code formats.

The authors curate a dataset of planning tasks presented in three formats and fine tune llms with four post-training methods: SFT, distillation, STaR, and GRPO(RL). They find that LLMs do not directly generalize across representations, and scaling doesn't solve this (in fact makes it a bit worse).

They then propose a two-stage curriculum: 1) training on symbolic data, then 2) on natural language which outperforms the single-representation training even with less NL data.

Finally, they analyze why this curriculum works, finding that successful generalization is better explained by structural similarity to training instances than by data frequency, meaning the curriculum encourages a form of generative analogical reasoning.

**Compliance With Llm Reviewing Policy:**

Affirmed.

**Key Questions For Authors:**

Q1: Suggestion: perhaps showing pre-ft performance in fig 2 might help? (or showing diffs)
Q2: This is a suggestion: perhaps swapping rows or cols to keep the in-distribution diagonal might be more intuitive for Table2.
Q3: I think the effect size of the analogy vs frequency analysis is a bit weak. But I might be misunderstanding how these metrics generally behave. Could the authors elaborate on how to interpret these results?

**Limitations:**

No.

Could the authors be clear about the broader impact?

**Strengths And Weaknesses:**

Soundness:
S: This is a well framed, executed and analyzed research. The question is an important one and the experiments are thorough. The analysis breaking down analogy vs frequency driven generalization is interesting.
W: The effect size of the analogy vs frequency analysis seems pretty weak.

Presentation:
S: The paper is mostly well presented.
W: Perhaps baselines can be added to figure 2.

Significance:
S: This provides direct actionable insights on how to augment the fine tuning data when a specific procedure is in mind.
W: However, its unclear how far the results generalize beyond the graph task tested here.

Originality:
S: The cross generalization of fine tuning is a relatively established question but this seems like a really good execution of that question in a nice formulation. The analogy vs frequency driven analysis is very interesting.

---

> ### Author Rebuttal · Authors · 2026-03-30
>
> We thank the reviewer for acknowledging the soundness and originality of the research question framing, execution, and analysis, as well as the clear presentation and significance of the research.
>
> To address your concerns:
>
> **Q1 & Q2 & W2**:  *Suggestion: perhaps showing pre-ft performance in fig 2 might help? (or showing diffs); This is a suggestion: perhaps swapping rows or cols to keep the in-distribution diagonal might be more intuitive for Table2.*
>
>
> **Response**: Thank you for the suggestions! We will update accordingly in the final version.
>
>
> **Q3 & W1**: *I think the effect size of the analogy vs frequency analysis is a bit weak… Could the authors elaborate on how to interpret these results?*
>
>
> **Response**: We appreciate the reviewer’s concern regarding the strength of the analogy-based correlation. Please note that we do not claim that analogical similarity fully explains model behavior, but rather that it provides a stronger account than frequency-based explanations (Section 5.1 in the main paper). This relative claim is directly supported by our experimental results, which consistently show stronger alignment with the analogy-based metric than with frequency baselines (Table 3 in the main paper). While it is true that the effect size is only moderate, this is expected since we are not considering many other sources of variance (e.g., surface form, model stochasticity, and representation complexity), and our analysis isolates the underlying procedure as a single explanatory factor. We will highlight this better in the final version of the paper.
>
> **W3**: *…its unclear how far the results generalize beyond the graph task tested here.*
>
>
> **Response**: We thank the reviewer for asking about cross-task generalization. While we focus on asynchronous planning in the main experiment for presentation clarity, we supply additional results in Section 4.3 and Table 2 on page 6, where we show that our conclusion is generalizable in Math and Physics. We will make these points clearer in the final version.
>
>
> To address your comment about the limitation:
>
>
> *Could the authors be clear about the broader impact?*
>
>
> **Response**: We thank the reviewer for their question about the limitation. As we discuss in the final impact statement section, this paper contributes to the understanding of LLMs’ generalization mechanism in terms of structured data, which is especially interesting at a time when LLMs are heavily trained and tested in symbolic domains such as code and graphs. Moreover, we examine the LLMs’ generalization capability in solving tasks presented in a non-standardized English dialect, and find that our proposed curriculum results in better generalization. However, we also notice that there is still a sensible gap in LLM performance between standardized and non-standardized English tasks. This indicates that better solutions can be proposed to mitigate technological unfairness. Last, we also note that LLMs still fall short of cross-representation generalization: while humans can often generalize in few- or even zero-shot, LLMs still need extensive in-representation training. We will clarify further in the final version.
>
>
> We deeply appreciate the reviewer’s efforts in giving feedback to help us improve the paper, and we are happy to discuss further!

---

> > ### Author Rebuttal · Reviewer_eL9F · 2026-04-02
> >
> > My concerns (which were minor) has been resolved.
> >
> > I think this is a good paper, I recommend for acceptance.
> >
> > The current (positive) score seems adequate.

---

### Decision · Program_Chairs · 2026-04-30

**Decision:**

Accept (regular)

**Comment:**

This paper investigates whether LLMs can transfer procedural knowledge across isomorphic representations (natural language, graph, code) of the same underlying task, using asynchronous planning as the primary testbed. The main finding is that symbolic post-training does not transfer zero-shot to natural language, but a two-stage curriculum, first training on graph representations, then continuing with GRPO on natural language, significantly outperforms training on natural language only. The paper further argues that successful generalisation is better explained by structural analogy than by frequency-based accounts.

There is much to like about this paper. The experimental design is well-conceived: by holding the underlying DAG procedure fixed and varying only the surface representation, the paper cleanly isolates the question of cross-representation transfer. The negative result (that symbolic training alone does not transfer to natural language) is valuable and clearly demonstrated. The proposed curriculum is simple yet effective, and the finding that GRPO is necessary (SFT and LoRA variants fail) offers actionable insights.
The analogy-based analysis, while not definitive, provides an interesting mechanistic lens that could open new avenues for future work. Results are confirmed across multiple model families (Qwen, Llama, OLMo) and supplemented with experiments on Math, Physics, and NL-AAVE, lending breadth to the conclusions. Finally, the rebuttal added substantial new evidence that strengthened the empirical picture considerably.

The paper does have some limitations. The primary testbed is an asynchronous planning task (results on maths and physics are provided but are less developed), and the analogy-based explanations are suggestive, rather than conclusive. Nonetheless, the paper asks an interesting question, provides a clean experiment and produces a non-obvious result (that curriculum order and RL-based exploration matter more than mere data mixing), which is useful for the field. The new experiments produced during the rebuttal further strengthen confidence in the findings. The authors are encouraged to include these results in an updated version of the paper